# Identification of Antimycobacterial Natural Products from a Library of Marine Invertebrate Extracts

**DOI:** 10.3390/medicines9020009

**Published:** 2022-01-28

**Authors:** Kojo Sekyi Acquah, Denzil R. Beukes, Ronnett Seldon, Audrey Jordaan, Suthananda N. Sunassee, Digby F. Warner, David W. Gammon

**Affiliations:** 1Department of Chemistry, University of Cape Town, Cape Town 7701, South Africa; acqkoj001@myuct.ac.za (K.S.A.); snsunassee@gmail.com (S.N.S.); 2School of Pharmacy, University of the Western Cape, Bellville 7535, South Africa; 3Drug Discovery and Development Centre, Department of Chemistry, University of Cape Town, Cape Town 7700, South Africa; ronnett.seldon@uct.ac.za; 4SAMRC/NHLS/UCT Molecular Mycobacteriology Research Unit & DST/NRF Centre of Excellence for Biomedical TB Research, Department of Pathology, Faculty of Health Sciences, University of Cape Town, Cape Town 7925, South Africa; audrey.jordaan@uct.ac.za (A.J.); digby.warner@uct.ac.za (D.F.W.); 5Institute of Infectious Disease and Molecular Medicine, Faculty of Health Sciences, University of Cape Town, Cape Town 7925, South Africa; 6Wellcome Centre for Infectious Diseases Research in Africa, University of Cape Town, Cape Town 7701, South Africa

**Keywords:** antitubercular, drug discovery, marine natural product, molecular networking, heteronemin, bengamides

## Abstract

Tuberculosis (TB) remains a public health crisis, requiring the urgent identification of new anti-mycobacterial drugs. We screened several organic and aqueous marine invertebrate extracts for their in vitro inhibitory activity against the causative organism, *Mycobacterium tuberculosis*. Here, we report the results obtained for 54 marine invertebrate extracts. The chemical components of two of the extracts were dereplicated, using ^1^H NMR and HR-LCMS with GNPS molecular networking, and these extracts were further subjected to an activity-guided isolation process to purify the bioactive components. *Hyrtios reticulatus* yielded heteronemin **1** and *Jaspis splendens* was found to produce the bengamide class of compounds, of which bengamides P **2** and Q **3** were isolated, while a new derivative, bengamide S **5**, was putatively identified and its structure predicted, based on the similarity of its MS/MS fragmentation pattern to those of other bengamides. The isolated bioactive metabolites and semi-pure fractions exhibited *M. tuberculosis* growth inhibitory activity, in the range <0.24 to 62.50 µg/mL. This study establishes the bengamides as potent antitubercular compounds, with the first report of whole-cell antitubercular activity of bengamides P **2** and Q **3**.

## 1. Introduction

Tuberculosis (TB), which is caused by *Mycobacterium tuberculosis* (Mtb), remains a leading cause of death, globally. Despite the availability of an effective anti-TB chemotherapy and a neonatal vaccine, an estimated 10 million people contracted TB and 1.4 million people died of the disease, globally, in 2019 [1]. The current standard combination therapy carries a high pill burden (up to four separate anti-TB drugs are used), long duration (minimum six months) of treatment and side effects or toxicities, owing to concomitant use of mixtures of TB drugs and non-TB drugs (for example, anti-HIV drugs) [1,2]. Furthermore, delayed diagnosis, inappropriate treatment and non-adherence has led to the emergence of multidrug-resistant (MDR) and extensively drug-resistant (XDR) Mtb strains [1,3]. Treatment of these resistant strains requires the use of non-frontline drugs, often over prolonged periods and with associated risks, including severe side effects and high therapeutic cost [3,4,5]. There is, therefore, an urgent need to discover and develop new potent anti-TB drugs with reduced side effects and effectiveness over shorter periods.

Nature remains our primary source of medication, with the marine environment relatively less explored [6]. The ocean occupies over 70% of the Earth’s surface and is uniquely rich in biodiversity. For their survival and existence, marine organisms, especially invertebrates, biosynthesize and secrete secondary metabolites to ward off competitors, predators and parasites [7]. These secondary metabolites have unique structural diversity, with a high degree of stereochemical complexity, and have shown a range of bioactivities, including anticancer, antioxidant, antifungal, antiviral, antibacterial, anti-TB, anti-inflammatory, antimalarial, analgesic and antinematodal properties [8,9]. Representative examples of marine-derived drugs include ziconotide, a peptide analgesic, derived from the marine cone snail *Conus magus*, and ecteinascidin-743, an alkaloid used in cancer treatment, derived from the sea squirt, *Ecteinascidia turbinate* [10,11]. Marine-natural products have promise in providing drug leads for the TB drug discovery pipeline and many anti-mycobacterial compounds, including the manzamines, halicyclamines, cyclostellettamines, batzelladines and callyaerins have been isolated from marine organisms [12].

Natural product drug discovery starts with the screening of natural product extracts as the preliminary step to finding new lead compounds. As part of our continuous search for new anti-TB agents from marine invertebrates, we screened several marine-natural product extracts, obtained from the United States of America’s National Cancer Institute (NCI) [13]. The whole-cell bioassay approach was employed, in preference to target-based and genetics-driven approaches, given the unknown bioactive compositions of the extracts and the well-described difficulties in ensuring whole-cell activity against Mtb [14]. Herein, we report the initial screening of 984 marine invertebrate extracts and the antimycobacterial activity results of 54 active extracts, with the detection and isolation of the active components of two of the extracts. The extracts of the sponges, *Hyrtios reticulatus* and *Jaspis splendens*, were among the most active (Table 1) and were subjected to activity-guided fractionation to isolate the active ingredients. Anti-mycobacterial activity in the extract from *Hyrtios reticulatus* was found to be associated with heteronemin **1**, while, as expected [15], the extract from *Jaspis splendens* produced the bengamide class of compounds (see Figure 1), with antitubercular activity shown, for the first time, to be associated with bengamides P and Q (Figure 1, compounds **2** and **3**), as well as other inseparable mixtures of bengamides. While other marine-derived and synthetically modified bengamides have previously been shown to be potent antitubercular agents and investigated as drug leads [16,17], to the best of our knowledge, this is the first report of antitubercular activity for bengamides P **2** and Q **3**.

## 2. Results and Discussion

### 2.1. Screening of Marine Invertebrate Samples

An initial set of 984 organic and aqueous marine invertebrate extracts were screened for their in vitro inhibitory activity against a fully virulent *M. tuberculosis* H37Rv fluorescent reporter mutant [16,18,19] and the minimum inhibitory concentrations required to inhibit 90% (MIC_90_) and 99% (MIC_99_) of the mycobacterial population were recorded. Of the 984 samples, 54 putative actives were identified and retested to confirm activity. A SciFinder^®^ (https://scifinder.cas.org/, accessed on 10 February 2021) search of the species of marine invertebrates, from which the 54 extracts were obtained, revealed that this was the first report of antimycobacterial activity in 44% of the species.

The 54 active extracts were from marine invertebrate samples, collected from the territorial waters of Mauritius (Indian Ocean) (2), South Africa (Indian and South Atlantic Oceans) (17), Tanzania (Indian Ocean) (8), Palau Island (Pacific Ocean) (4), and Papua- New Guinea (Pacific Ocean) (23). The invertebrates are of the phyla Porifera (46), Cnidaria (2), Chordata (4) and Mollusca (2). The actives displayed MIC, in the range of <0.24 to 125 µg/mL (Table 1), with 35% exhibiting good activity (MIC < 25 µg/mL), 28% moderate activity (MIC 25–65 µg/mL) and 37% weak activity (MIC 65–125 µg/mL).

The extracts of the species belonging to the genera *Jaspis*, *Didemnum*, *Acanthochitona*, *Isodictya*, *Hemitedania*, *Neopetrosia*, *Aplysinopsis*, *Cypraea*, *Lissoclinum*, *Fascaplysinopsis*, *Ircinia* and *Agelas*, exhibited the most potent activities, with sub-10 ug/mL MICs. It is worth noting that the SciFinder^®^ search for these genera returned no reports of antimycobacterial activity, except for the genera, *Didemnum*, *Neopetrosia*, *Lissoclinum*, *Ircinia* and *Agelas*. The diterpene alkaloids, agelasines, isolated from the genus *Agelas*, and the tetracyclic bis-piperidine alkaloid, neopetrosiamine A, isolated from *Neopetrosia proxima*, as well as crude extracts of species of the genera *Didemnum*, *Lissoclinum*, *Ircinia* and *Agelas*, have been reported to show antimycobacterial activities [20,21,22,23]. This reaffirms that marine invertebrates represent a potential source of novel anti-TB compounds.

### 2.2. Dereplication, Molecular Networking, Isolation and Structure Elucidation

The extracts from the two Mauritian marine sponges *Hyrtios reticulatus* (SS10, MIC_90_ = 2.40 μg/mL) and *Jaspis splendens* (SS2, MIC_90_ = 51.00 μg/mL) were selected for preliminary investigations, in an attempt to isolate and identify the active compounds. Both extracts were available in appreciable amounts and were examples of extracts with good and moderate MIC_50_ values, respectively. The actives were isolated using an activity-guided isolation procedure. First, the ^1^H NMR spectra and HR-LCMS profiles were obtained for the bioactive extracts, and these were then subjected to bioactivity-guided fractionation, using a normal-phase solid-phase extraction (SPE) procedure, which, in each case, yielded five fractions (A–E), corresponding to polarities of the eluents. The fractions were then screened against the Mtb H37Rv reporter strain.

The HR-LCMS profile of the extract of *Hyrtios reticulatus* (SS10) showed a major peak, with *m*/*z* 511.3026 [M + Na]^+^, which was identified as the antimycobacterial sesquiterpene, heteronemin **1**; this was confirmed by the ^1^H NMR spectrum. Fractions A and B of SS10 were the most active but had the same ^1^H NMR spectrum and were, therefore, combined and further purified, using normal-phase column chromatography to yield the known compound, heteronemin **1**, as the most active ingredient. Heteronemin **1** was isolated as a white amorphous solid, with a molecular formula of C_29_H_44_O_6_, deduced from HRESIMS (observed [M + Na]^+^ = 511.3026; calculated [M + Na]^+^ = 511.3036; ∆ = −1.96 ppm), corresponding to eight degrees of unsaturation (Appendix A). The structure of compound **1** was fully elucidated by the analysis of HRESIMS, together with 1D and 2D NMR data, and this was congruent with that reported in the literature (Appendix A) [24,25].

The HR-LCMS/MS data of the extract of *Jaspis splendens* (SS2) was analyzed on the GNPS molecular networking platform [26]. Dereplication of the nodes showed molecular network families, which are representative of the bengamide class of compounds; this was confirmed by the analysis of the ^1^H NMR spectrum of SS2 (Figure 2, Appendix A). The bengamides are a class of compounds with a core scaffold, comprising a 2(R)-methoxy-3(R),4(S),5(R)-trihydroxy-8-methylnon-6(E)-enoyl moiety, linked to an aminocaprolactam with its cyclic amide, nitrogen free or methylated [15,16,27]. Approximately 23 bengamides have been isolated, with various structural variations, enabling grouping into structural classes [16,27]. The first type contains a hydroxylysine-derived caprolactam, with the OH at C-13 either unsubstituted (bengamides Y, Z) or acylated by a lipid unit (bengamides A, B, G–J, L–O), or by polyketide esters (bengamides C, D). The second type has the lysine-derived caprolactam (which therefore does not have an OH at C-13), but with the 5-OH acylated by the lipid unit (bengamides P–R), or not (bengamides E, E′, F, F′) [16,27]. Some bengamides have the same molecular formulae and weight and are only differentiated by characteristic NMR signals. For example, bengamides J and M have the same molecular formula of C_33_H_60_N_2_O_8_ and weight 635.4228 [M + Na]^+^ (Figure 2, Appendix A) [27].

Molecular cluster F1 contains the first class of bengamides with a lipid chain (Figure 2). The node with *m*/*z* 621.408 [M + Na]^+^ corresponds to isomeric bengamides B **9**, I **16**, L **18** or O **21**, while the node with *m*/*z* 607.392 [M + Na]^+^ corresponds to bengamides A **8**, H **15** or N **20**, and the node with *m*/*z* 635.424 [M + Na]^+^ to bengamides J **17** or M **19** (Figure 1 and Figure 2). Family F2, which is constituted by four nodes, was dereplicated to be the second type of bengamides with a fatty acid chain, including bengamides P **2** (*m*/*z* 591.397 [M + Na]^+^), Q **3** (*m*/*z* 605.413 [M + Na]^+^) and R **4** (*m*/*z* 619.428 [M + Na]^+^), which was linked to its new analogue **5** (node with *m*/*z* 633.445 [M + Na]^+^), with a mass difference of 14 Da, signifying methylation. From the biosynthesis of bengamides and a keen examination of isolated bengamide structures, the methylation would either be on the caprolactam ring nitrogen or an elongation of the lipid chain of **4**. Evaluation of the fragmentation patterns (Appendix A) of compounds **4** and **5**, showed that **5** is an N-methylated analogue of **4**. Compound **5** was therefore assigned the name bengamide S. Fractions B and C of SS2 exhibited the best activities, with potent MIC_90_ values of <0.24 µg/mL and 0.25 µg/mL, respectively. The HR-LCMS profiles and ^1^H NMR spectra of active fractions B and C, revealed that they comprised similar mixtures of the bengamides, and these fractions were, therefore, combined and subjected to normal-phase column chromatography. From >150 fractions collected, fractions F94 and F100 were sufficiently homogeneous for their structures to be elucidated as compounds **2** and **3**, respectively. Compounds **2** and **3** had ^1^H NMR signals characteristic of the second type of bengamides, with a lipid chain located as esters of the C-5 oxygen. The fact that this was attached to C-5 and not C-13 was evident from the downfield shift of the signal at δ_H_ 5.47, for the methine proton H-5 in the ^1^H NMR spectrum, compared to the shielded signals for the diastereotopic C-13 methylene protons at δ_H_ 1.88 and 1.43. The structures of compounds **2** and **3** were fully elucidated by analyses of their HR-ESIMS, 1D and 2D NMR data and comparison with the literature, and confirmed to be bengamides P and Q, respectively (Appendix A) [27].

The ^1^H NMR spectra of active fractions F107, F114 and F130 (Appendix A) showed characteristic signals for bengamides of both classes with a lipid chain. However, the main component of fraction F130 was the first type of bengamide, with a branched lipid chain and characteristic upfield signals in the ^1^H NMR spectrum at δ_H_ 0.7–1.7, and a signal for an N-methyl group in the caprolactam ring (δ_H_ 3.13). It is proposed that this major component may be bengamide M, as a corresponding peak of *m*/*z* 635.423 [M + Na]^+^ was identified in the HR-LCMS profile of fraction F130.

### 2.3. Antimycobacterial Activity of Isolated Compounds and Semi-Pure Fractions

Heteronemin **1** exhibited an MIC_90_ of 8.23 µg/mL against Mtb. Shown in Table 2 are the antimycobacterial activity results of isolated pure compounds and the most active semi-pure fractions obtained from purification of fractions B and C of SS2. Fraction F114 showed the greatest activity, with MIC_90_ < 0.24 µg/mL. Although bengamides have been isolated from the genus *Jaspis*, this is the first report of the detection and isolation of bengamides from *Jaspis splendens*, which is known to be a prolific producer of jasplakinolides [16].

The bengamides are potent antitumor agents, which target methionine aminopeptidase (MetAP) [15,16]. Synthetic MetAP inhibitors, based on the bengamide structural scaffold, have been described with activity against the Mtb MetAPs (the bacillus encodes two isoforms, MetAP1a and MetAP1c); however, the antimycobacterial potential (whole-cell or target-based) of natural bengamides is less explored [28,29]. There is a report of antimycobacterial activity of bengamide B **9**, which exhibited strong in vitro activity against whole-cell Mtb and the purified Mtb MetAP1C protein and was also non-toxic against human cell lines [17]. To our knowledge, ours is the first report of antimycobacterial activity for the second type of bengamides with lipid chains, namely bengamides P **2** and Q **3**. This suggests that the bengamides are potential antitubercular leads and that all known natural analogs could be screened against Mtb, to further aid in SAR and QSAR design and studies. There also appears to be value in further exploration of the Mtb MetAPs as potential drug targets.

## 3. Materials and Methods

### 3.1. General Experimental Procedure

NMR spectra were obtained on a Bruker Ascend 600 (Bruker, Billerica, MA, USA) cryoprobe prodigy at 600 MHz and 150 MHz for ^1^H and ^13^C nuclei, respectively. Chemical shifts were referenced using the corresponding undeuterated solvent signals in CD_3_OD (δ_H_ 3.30, δ_C_ 47.61) and CDCl_3_ (δ_H_ 7.25, δ_C_ 77.00). High-resolution mass spectrometric data were obtained using a Thermo Instrument MS system (LTQ XL/LTQ Orbitrap Discovery, Thermo Scientific, Bremen, Germany) coupled to a Thermo Instrument HPLC system (Accela PDA detector, Accela PDA autosampler and Accela pump, Thermo Scientific, Bremen, Germany). The following conditions were used: capillary voltage 45 V, capillary temperature 260 °C, auxillary gas flow rate 10–20 arbitrary units, sheath gas flow rate 40–50 arbitrary units, spray voltage 4.5 kV, mass range 100–2000 amu (maximum resolution 30,000). All solvents used throughout were HPLC-grade and purchased from both Merck and Sigma-Aldrich. Column chromatography was carried out on silica gel 60 (Fluka 70–230 mesh, 63–200 μm, Sigma-Aldrich, Buchs, Switzerland), and preparative TLC on silica gel 60 Analtech GF254 (20 × 20 cm, 2000 μm, Analtech Inc., Newark, DE, USA). Analytical TLC was performed on Merck silica gel 60 F254 (Merck KGaA, Darmstadt, Germany) and silica gel 60 RP-18 F254 plates and bands were visualized based on the UV absorbance at 254 nm and by heating after staining with ceric ammonium sulfate reagent.

### 3.2. Marine Invertebrate Samples

The screening consisted of 984 organic (MeOH-CH_2_Cl_2_, 1:1) and aqueous (H_2_O) extracts of marine invertebrate samples obtained from the National Cancer Institute (NCI), USA [13].

### 3.3. Antimycobacterial Activity

The crude extracts, semi-pure and isolated compounds were dissolved in DMSO and their minimum inhibition concentrations (MIC_90_/_99_ values) determined against the green fluorescent protein (GFP)-tagged Mtb H37Rv pMSp12:GFP bioreporter using the standard broth microdilution method developed by Collins et al. [18]. Mtb H37Rv was cultured in Middlebrook 7H9 broth medium supplemented with albumin–dextrose complex, D-glucose, and Tween-80 (7H9/ADC/Glu/Tw). The minimum concentration which inhibited the growth of 90% (MIC_90_) or 99% (MC99) of bacilli for the tested samples was determined at 1 week and 2 weeks post-inoculation using microplate detection of GFP fluorescence intensity and expressed in µg/mL. The fluorescence was measured with excitation at 485 nm and emission at 520 nm. Growth media and 5% dimethyl sulfoxide (DMSO) were used as a negative control, with rifampicin used as a positive control.

### 3.4. Fractionation, Isolation and Purification of Active Compounds

The extracts SS2 and SS10 were subjected to an NCI DIOL SPE fractionation process. The organic extract (150 mg) was weighed and solubilized in 1.8 mL of 1:1 MeOH/CH_2_Cl_2_ and sonicated to mix. The mixture was loaded on a 2 g DIOL SPE column and eluted stepwise, with solvent mixtures of increasing polarity (a column volume/solvent mixture of 6 mL): 9:1 hexane/CH_2_Cl_2_ (A), 20:1 CH_2_Cl_2_/EtOAc (B), 100% EtOAc (C), 5:1 EtOAc:MeOH (D), 100% MeOH (E). The fractionation process was repeated with 3 × 150 mg of extract and each fraction, from A to E, was combined and dried.

Fractions A and B of SS10 were combined and further purified through a normal-phase column chromatography with an increasing polarity of a mixture of hexane/EtOAc from 1:0 hexane/EtOAc to 0:1 hexane/EtOAC to yield compound **1** (17.01 mg).

Fractions B and C of SS2 were also combined and subjected to a normal-phase column chromatography with an increasing polarity of a mixture of hexane/EtOAc from 1:0 hexane/EtOAc to 0:1 hexane/EtOAc to yield compounds **2** (3.08 mg) and **3** (0.69 mg) and a mixture of bengamides.

Heteronemin **1**: white amorphous solid; for ^1^H, ^13^C NMR data, see Appendix A; HRESIMS (positive mode) *m*/*z* 511.3026 [M + Na]^+^ Δ −1.96 ppm; calculated for C_29_H_44_O_6_.

Bengamide P **2**: colourless oil; for ^1^H, ^13^C NMR data, see Appendix A; HRESIMS (positive mode) *m*/*z* 591.3978 [M + Na]^+^ Δ 2.03 ppm; calculated for C_31_H_56_N_2_O_7_.

Bengamide Q **3**: colourless oil; for ^1^H, ^13^C NMR data, see Appendix A; HRESIMS (positive mode) *m*/*z* 605.4133 [M + Na]^+^ Δ 1.82 ppm; calculated for C_32_H_58_N_2_O_7_.

### 3.5. Dereplication and Molecular Networking

Raw LC–MS/MS data of sample SS2 was converted to mzXML format using the ProteoWizard tool MSconvert (version 3.0.10051, Vanderbilt University, United States) [30]. The mzXML data was uploaded to the Global Natural Products Social (GNPS) Molecular Networking (MN) webserver3 (http://gnps.ucsd.edu, accessed on 12 August 2020) and analyzed using the MN workflow [23]. The data was filtered by removing all MS/MS fragment ions within +/− 17 Da of the precursor *m*/*z*. MS/MS spectra were window filtered by choosing only the top 6 fragment ions in the +/− 50Da window throughout the spectrum. The precursor ion mass tolerance was set to 0.02 Da and a MS/MS fragment ion tolerance of 0.02 Da. A network was then created where edges were filtered to have a cosine score above 0.7 and more than 3 matched peaks. Further, edges between two nodes were kept in the network, only if each of the nodes appeared in each other’s respective top 10 most similar nodes. Finally, the maximum size of a molecular family was set to 100, and the lowest scoring edges were removed from molecular families until the molecular family size was below this threshold. The spectra in the network were then searched against GNPS’ spectral libraries. The library spectra were filtered in the same manner as the input data. All matches kept between network spectra and library spectra were required to have a score above 0.7 and at least 3 matched peaks. The output of the molecular network was visualized using Cytoscape version 3.7.2 [31] and displayed using the settings “preferred layout” with “directed” style. The nodes (compounds) originating from the solvent control (MeOH) were excluded from the original network to enable visualization of only the compounds in SS2.

## 4. Conclusions

Marine-natural products are a reliable source of potent antitubercular leads, as this screening project identified 54 actives in whole-cell growth inhibition assays (MIC_90_ range: <0.244–125 µg/mL). Notably, 44% of the species from which these extracts were obtained are reported here, for the first time, as possessing antimycobacterial activity. A combination of ^1^H NMR and HR-LCMS dereplication, along with GNPS molecular networking and a bioactivity-guided isolation techniques was employed to detect and isolate heteronemin and the bengamides P and Q, as the respective active ingredients of the extracts of the Mauritian sponges *Hyrtios reticulatus* and *Jaspis splendens*. A new bengamide derivative was detected in the molecular network of SS2. In this study, the bengamides have been identified as potent antimycobacterial compounds and should be explored further. Moreover, the isolation of promising active compounds from crude extracts, with high and moderate activity, provides compelling evidence for the continuous need to explore the rich resource of the NCI’s repository of marine invertebrate extracts further, for novel anti-TB leads.

## Figures and Tables

**Figure 1 medicines-09-00009-f001:**
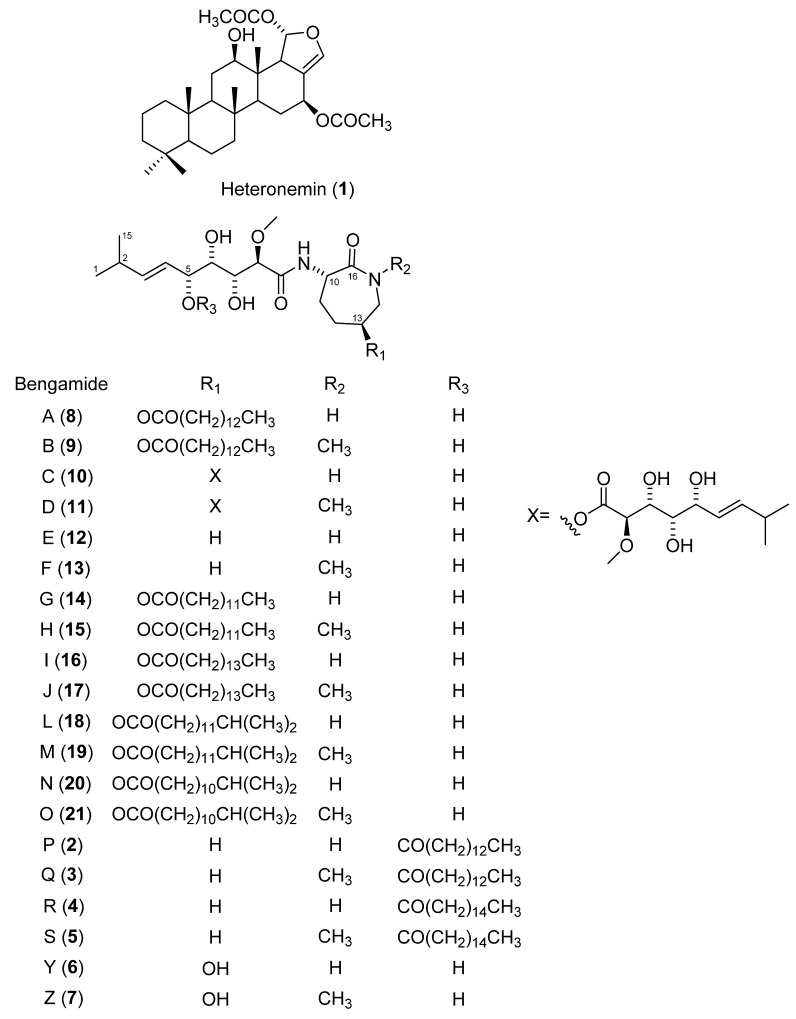
Structures of isolated compounds (**1–3**), predicted (**5**) and some known bengamides (**4**, **6–21**).

**Figure 2 medicines-09-00009-f002:**
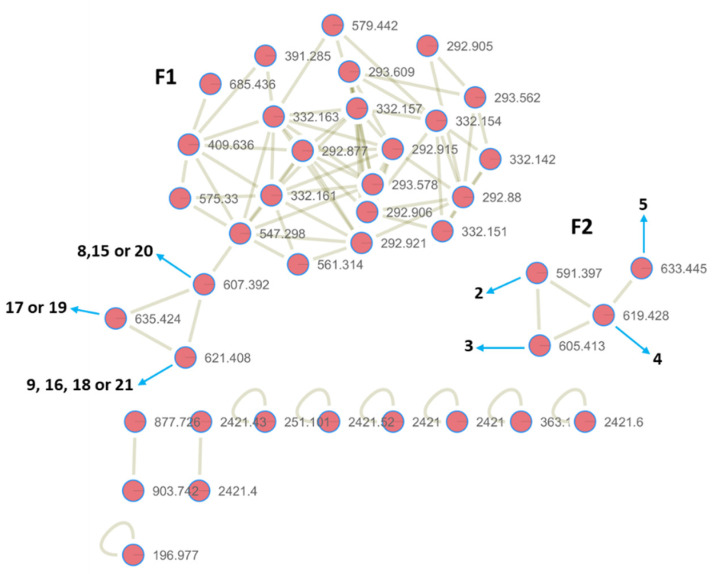
GNPS molecular network of the crude extract (SS2) of marine sponge *Jaspis splendens* containing some annotated nodes. Numbers in bold correspond to compounds shown in Figure 1.

**Table 1 medicines-09-00009-t001:** Minimum inhibitory concentration (MIC_90_ and MIC_99_) of marine invertebrate extracts against *M. tuberculosis* H37Rv and classification to species level of the marine invertebrates with their source country.

Country	Phylum	Class	Order	Family	Genus	Species	NPID	MIC_90_ (μg/mL)	MIC_99_ (μg/mL)
Mauritius	Porifera	Demospongiae	Astrophorida	Ancorinidae	*Jaspis*	*splendens (Dorypleres)*	C019765	2.4	4.5
	Porifera	Demospongiae	Dictyoceratida	Thorectidae	*Hyrtios*	*reticulatus*	C019725	51.0	72.0
South Africa	Porifera	Demospongiae	Poecilosclerida	Guitarridae	*Guitarra*	*fimbriata indica*	C018520	70.0	87.0
	Porifera	Demospongiae	Hadromerida	Suberitidae	*Aaptos*	sp. 1	C018442	71.0	83.0
	Porifera	Demospongiae	Verongida	Aplysinellidae	*Porphyria*	*pedunculata*	C018530	71.0	94.0
	Porifera	Demospongiae	Poecilosclerida	Hymedesmiidae	*Phorbas*	sp. 1	C018462	72.0	81.0
	Porifera	Demospongiae	Halichondrida	Heteroxyidae	*Higginsia*	*bidentifera*	C018466	72.0	83.0
	Porifera	Demospongiae	Halichondrida	Heteroxyidae	*Higginsia*	sp. 1	C018578	72.0	84.0
	Cnidaria	Hydrozoa	Leptothecatae	Aglaopheniidae	*Lytocarpia*	*formosa*	C018502	73.0	81.0
	Porifera	Demospongiae	Halichondrida	Halichondriidae	*Hymeniacidon*	sp. 1	C018470	73.0	84.0
	Porifera	Demospongiae	Poecilosclerida	Guitarridae	*Guitarra*	*fimbriata indica*	C018458	73.0	91.0
	Chordata	Ascidiacea	Aplousobranchia	Didemnidae	*Didemnum*	*obscurum*	C019906	1.6	1.7
	Porifera	Demospongiae	Halichondrida	Dictyonellidae	*Stylissa*	sp. 1, n.sp.	C019904	17.0	19.4
	Cnidaria	Anthozoa	Alcyonacea	Nephtheidae	*Drifa*	sp. b (n.sp.)	C018631	41.0	52.0
	Porifera	Demospongiae	Poecilosclerida	Raspailiidae	*Echinodictyum*	sp. 1	C018566	25.1	42.8
	Mollusca	Polyplacophora	Chitonida	Acanthochitonidae	*Acanthochitona*	*garnoti*	C018637	0.5	0.63
	Porifera	Demospongiae	Poecilosclerida	Isodictyidae	*Isodictya*	sp. 2, n.sp.	C018626	18.8	6.25
	Porifera	Demospongiae	Poecilosclerida	Tedaniidae	*Hemitedania*	sp. 1	C018538	18.7	9.3
	Porifera	Demospongiae	Halichondrida	Halichondriidae	*Halichondria*	sp. 1	C018460	13.8	12.2
Tanzania	Porifera	Demospongiae	Haplosclerida	Petrosiidae	*Neopetrosia*	*tuberosa* cf.	C015405	2.5	3.2
	Porifera	Demospongiae	Haplosclerida	Phloeodictyidae	*Oceanapia*	*ramsayi*	C015331	22.0	28.0
	Porifera	Demospongiae	Haplosclerida	Phloeodictyidae	*Oceanapia*	sp. 3	C015461	35.0	42.0
	Porifera	Demospongiae	Poecilosclerida	Chondropsidae	*Chondropsis*	sp. 1	C015479	58.0	64.0
	Chordata	Ascidiacea	Aplousobranchia	Polycitoridae	*Eudistoma*	*giganteum*	C015357	61.0	68.0
	Porifera	Demospongiae	Dictyoceratida	Thorectidae	*Aplysinopsis*	*elegans* cf.	C015228	2.5	2.60
	Mollusca	Gastropoda	Neogastropoda	Cypraeidae	*Cypraea*	*tigris*	C015272	47.9	57.9
	Porifera	Demospongiae	Poecilosclerida	Podospongiidae	*Diacarnus*	*ardoukobae*	C015180	70.8	81.2
Papua-New Guinea	Chordata	Ascidiacea	Aplousobranchia	Didemnidae	*Lissoclinum*	*badium*	C018795	2.6	3.1
	Porifera	Demospongiae	Dictyoceratida	Thorectidae	*Fascaplysinopsis*	sp. 2	C018781	7.9	16.5
	Porifera	Demospongiae	Haplosclerida	Petrosiidae	*Petrosia (Strongylophora)*	*corticata*	C018743	31.0	35.0
	Porifera	Demospongiae	Dictyoceratida	Dysideidae	*Dysidea*	sp. 17	C018675	32.0	36.0
	Porifera	Demospongiae	Haplosclerida	Chalinidae	*Haliclona (Gellius)*	sp. 4	C018717	36.0	57.0
	Porifera	Demospongiae	Dictyoceratida	Dysideidae	*Lamellodysidea*	*herbacea*	C018667	41.0	43.0
	Porifera	Demospongiae	Dictyoceratida	Thorectidae	*Carteriospongia*	sp. 3	C018787	48.0	70.0
	Porifera	Demospongiae	Dictyoceratida	Thorectidae	*Phyllospongia*	sp. 7	C018595	63.0	70.0
	Porifera	Demospongiae	Dictyoceratida	Thorectidae	*Aplysinopsis*	*elegans*	C018651	65.0	71.0
	Porifera	Demospongiae	Poecilosclerida	Mycalidae	*Mycale*	*setosa*	C018783	72.0	81.0
	Porifera	Demospongiae	Dictyoceratida	Spongiidae	*Spongia*	sp. 5	C018671	72.0	82.0
	Chordata	Ascidiacea	Aplousobranchia	Polyclinidae	*Synoicum*	*castellatum*	C018601	72.0	83.0
	Porifera	Demospongiae	Haplosclerida	Petrosiidae	*Neopetrosia*	*tuberosa* cf.	C018593	72.0	83.0
	Porifera	Demospongiae	‘Lithistid’	Theonellidae	*Theonella*	sp. 7	C018729	73.0	83.0
	Porifera	Demospongiae	Dictyoceratida	Irciniidae	*Ircinia*	*arbuscula* cf.	C018773	< 0.24	0.5
	Porifera	Demospongiae	Hadromerida	Suberitidae	*Aaptos*	*nigra*	C018695	58.3	51.4
	Porifera	Demospongiae	Agelasida	Agelasidae	*Agelas*	*oxeata* cf.	C018756	0.4	0.9
	Porifera	Demospongiae	‘Lithistid’	Scleritodermidae	*Aciculites*	*oxtylota*	C018740	20.8	21.9
	Porifera	Demospongiae	‘Lithistid’	Scleritodermidae	*Microscleroderma*	*herdmani*	C018684	23.7	30.6
	Porifera	Demospongiae	Haplosclerida	Callyspongiidae	*Callyspongia (Euplacella)*	*elongata* cf.	C018714	85.2	125.0
	Porifera	Demospongiae	Haplosclerida	Niphatidae	*Pachychalina*	sp. 11	C018694	17.9	46.5
	Porifera	Demospongiae	Astrophorida	Thrombidae	*Thrombus*	sp. 1	C018716	20.8	42.5
	Porifera	Demospongiae	Haplosclerida	Niphatidae	*Niphates*	*elegans*	C018720	23.8	91.3
Palau Islands	Porifera	Demospongiae	Halichondrida	Halichondriidae	*Topsentia*	*cavernosa*	C019960	71.9	78.9
	Porifera	Demospongiae	Hadromerida	Suberitidae	*Aaptos*	*nigra*	C019928	85.2	106.0
	Porifera	Demospongiae	Haplosclerida	Chalinidae	*Haliclona (Gellius)*	sp. 4	C020034	95.9	120.0
	Porifera	Demospongiae	Verongida	Aplysinellidae	*Porphyria*	sp. 1, n.sp.	C019898	113.0	125.0
Control	Rifampicin							0.02	0.03

**Table 2 medicines-09-00009-t002:** Minimum inhibitory concentration of bengamides P **2**, Q **3** and semi-pure fractions, obtained from purification of fractions B and C of SS2, against Mtb H37Rv, cultured for 7 and 14 days.

Compound/Fraction	MIC_90_ (µg/mL): 7 Days	MIC_90_ (µg/mL): 14 Days
**2**	2.14	1.03
**3**	62.50	31.25
F107	1.09	0.74
F114	<0.24	<0.24
F130	0.65	0.49
Rifampicin	0.02	0.01

## Data Availability

Not applicable.

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
