# Peer review of "Identification of Antimycobacterial Natural Products from a Library of Marine Invertebrate Extracts"

_medicines, 2022, doi:10.3390/medicines9020009_

Round 1

Reviewer 1 Report

Manuscript number: medicines-1548047

Identification of antimycobacterial natural products from a library of marine invertebrateextracts
In this paper Gammon and co-workers present some interesting and innovative results on marine
invertebrate extracts and their in vitro inhibitory activity against Mycobacterium tuberculosis.

The work is well presented, and a lot of experimental work was done, nevertheless some revision
on the whole manuscript should be done, in order to help readers in understanding the topic, the
structure and the importance of the whole work.

In particular,

1. The introduction should require some additional information that are present later in the
manuscript.

I suggest author to anticipate all the data concerning literature knowledge on bengamide
chemistry, structures, and bioactivity (as reported later at page 10, lines 140-152 or page 12
lines 198-240). As a result, the novelty of this manuscript could be noticed from the
introduction.

I suggest also to include a first Figure, presenting known bengamides (especially P(2) and Q
(3)).

2. Table 1 is too long and not effective. I suggest author to find the best way to divide it into
some sub-tables, may dividing for the country, removing some classifications, or firstly
considering MICs. In a so long table, the most important data are completely lost.

3. I suggest also authors to change the way they call the Fractions (as at page 11, line 174). In
my opinion, a different numbering could be useful.

4. Considering the good results on M. tuberculosis, why the authors did not consider any other
mycobacteria or bacteria to be tested?

Author Response

  1. The introduction should require some additional information that are present later in the manuscript.
    I suggest author to anticipate all the data concerning literature knowledge on bengamide chemistry, structures, and bioactivity (as reported later at page 10, lines 140-152 or page 12
    lines 198-240). As a result, the novelty of this manuscript could be noticed from the introduction.
    I suggest also to include a first Figure, presenting known bengamides (especially P(2) and Q
    (3)).

These are useful suggestions. We have modified the last paragraph of the introduction to specifically refer to the bengamide structures and to draw attention, via citation here of the key review in ref [25] to the extensive knowledge already documented on these structures, but to also highlight our results of isolation of key bengamide structures and demonstrating for the first time that these have antitubercular properties. We feel it is not necessary to reproduce the structures at this point, but have (a) introduced a citation of Figure 1 at this point, which has a full summary of the structures, and (b) note that the structures are also already shown in the graphical abstract.

  1. Table 1 is too long and not effective. I suggest author to find the best way to divide it into some sub-tables, may dividing for the country, removing some classifications, or firstly considering MICs. In a so long table, the most important data are completely lost.

The concern is understood. However, we would prefer to keep Table 1 in its present form, noting for instance, that the Table already separates organisms and extracts according to country of origin. We believe the Table represents one of the core findings of this research, and while there is indeed a great deal of detail, it is necessary detail in order to report thoroughly and unambiguously on the biological origins of the extracts that have been examined.

If, however, the editors would prefer for editorial reasons to truncate the Table, we would consider only including the first two rows of data in the manuscript itself (since these are most relevant to the further results in the manuscript) and to place the rest of the Table in the Supplementary Information with appropriate citation in the text of the manuscript.

  1. I suggest also authors to change the way they call the Fractions (as at page 11, line 174). In my opinion, a different numbering could be useful.

The reviewer has not suggested a particular format for naming of fractions, and we are not aware of any standard requirements from the Journal in this respect. We have chosen to label the fractions in the format “F(number)”, as in “F94” or “F101” which we consider logical and simple: if the editors have a particular format that they prefer, we will take advice.

  1. Considering the good results on M. tuberculosis, why the authors did not consider any other mycobacteria or bacteria to be tested?

The question/suggestion is noted. The focus of this study and related work in our research group is on anti-tubercular activity and the search for promising sources of potential leads for new drugs to treat TB – hence our focus on M. tuberculosis. The work may indeed be extended to other bacteria later, but, as stated in our introduction, TB is a significant component of the health-care burden in our country and globally, and warrants a focus.

Reviewer 2 Report

Dears authors

Analysis by paper partitions:

1 - Introduction: the content and the drafting of the general part must be reformed to review the syntax of the topic

2- Discussion:

deepen in consideration of the problem of antifungal resistance the use of essential oils against multidrug-resistant strains of Mycobacteria and clinical applications. Learn more about this by using and citing the following references:  PMID: 30518254 ; PMID: 31657682 ; PMID: 33707371.

3 - Check the bibliographic entries throughout the text, some of which are non-compliant, review some entries in the references and necessarily insert those referred to in point 2 for the purpose of my acceptance.

4 - Review English grammar and in particular applied scientific English: in particular, tenses and syntax in the discussion.

Author Response

1 - Introduction: the content and the drafting of the general part must be reformed to review the syntax of the topic

The reviewer has not specified any corrections here: our view is that the language, syntax and scientific style of our manuscript is broadly within the scope of this journal. We note again that we have made some modifications to the text in line with the comments and suggestions of Reviewer 1.

2- Discussion:

deepen in consideration of the problem of antifungal resistance the use of essential oils against multidrug-resistant strains of Mycobacteria and clinical applications. Learn more about this by using and citing the following references:  PMID: 30518254 ; PMID: 31657682 ; PMID: 33707371.

Our view is that these papers cited by the Reviewer are NOT relevant to the specific scope of our study, in that they deal with antimycobacterial properties of essential oils from plant origins whereas the specific focus of our work is on marine organisms as potential sources of antimycobacterials. To incorporate these references would require significantly extending the scope of the review to cover an undoubtedly large body of literature on plant-derived natural products.

The first two of the recommended papers (PMID: 30518254 and PMID: 31657682) deal with anti-mycobacterial activity of the essential oils derived from the plants Melaleuca cajuputi (PMID: 30518254), and Trachyspermum copticum and Pelargonium graveolens (PMID: 31657682).  The third recommended paper (PMID: 33707371) deals with the formation of biofilms by nontuberculous mycobacteria (NTM) and investigation of (a) inhibition of these biofilms by plant essential oils, and (b) determination of antimycobacterial effect of essential oils by the semi-quantitative disc diffusion method. This is therefore also beyond the scope of our research and our manuscript, which is specifically focused on marine sources of known and novel antimycobacterials

( For reference, the relevant citations suggested by the Reviewer were as follows:

PMID: 30518254  Nat Prod Res, 2020 May;34(10):1494-1497. doi: 10.1080/14786419.2018.1509335. Epub 2018 Dec 5. "In vitro" activity of Melaleuca cajuputi against mycobacterial species 

PMID: 31657682  Recent Pat Antiinfect Drug Discov. 2020;15(1):68-74.

 doi: 10.2174/1574891X14666191028113321.

The Chemical Composition and Anti-mycobacterial Activities of Trachyspermum copticum and Pelargonium graveolens Essential Oils

PMID: 33707371  Int J Mycobacteriol, Jan-Mar 2021;10(1):43-50.

 doi: 10.4103/ijmy.ijmy_228_20.

The biofilm formation of nontuberculous mycobacteria and its inhibition by essential oils

3 - Check the bibliographic entries throughout the text, some of which are non-compliant, review some entries in the references and necessarily insert those referred to in point 2 for the purpose of my acceptance.

In the absence of specific details on this, we are not certain what is meant by “non-compliant”, and hence which references require modification and in which ways. If the final editorial process reveals corrections required we will certainly be happy to make these.

4 - Review English grammar and in particular applied scientific English: in particular, tenses and syntax in the discussion.

Again, in the absence of specific details, we are not sure what aspects need changing – and reiterate our view that our manuscript is broadly appropriate in respect of the scientific and grammatical style required by the journal.

Round 2

Reviewer 1 Report

I continue suggesting authors to divide the table, nevertheless the actual manuscript could be ready for publication.

Reviewer 2 Report

Been made the corrections.